# Drug-Induced Myoclonus: A Systematic Review

**DOI:** 10.3390/medicina61010131

**Published:** 2025-01-15

**Authors:** Jamir Pitton Rissardo, Ana Letícia Fornari Caprara, Nidhi Bhal, Rishikulya Repudi, Lea Zlatin, Ian M. Walker

**Affiliations:** 1Neurology Department, Cooper University Hospital, Camden, NJ 08103, USA; fornari-caprara-ana@cooperhealth.edu (A.L.F.C.); walker-ian@cooperhealth.edu (I.M.W.); 2Medicine Department, Jehangir Hospital, Pune 411001, India; drnidhiraut@gmail.com; 3Medicine Department, Apollo Institute of Medical Sciences and Research, Hyderabad 500072, India; rishikulyarepudi@gmail.com; 4Neuroscience Department, Ohio State University, Columbus, OH 43210, USA; leazlatin@gmail.com; 5Neurology Department, Cooper Medical School of Rowan University, Camden, NJ 08103, USA

**Keywords:** myoclonus, neurotoxicity, encephalopathy, drug-induced myoclonus, adverse effect, myoclonus/chemically induced, antidepressant-induced myoclonus, opioid-induced myoclonus, anti-seizure medication-induced myoclonus, antibiotic-induced myoclonus

## Abstract

*Background and Objectives*: Myoclonus is already associated with a wide variety of drugs and systemic conditions. As new components are discovered, more drugs are suspected of causing this disabling abnormal involuntary movement. This systematic review aims to assess the medications associated with drug-induced myoclonus (DIM). *Materials and Methods*: Two reviewers assessed the PubMed database using the search term “myoclonus”, without language restriction, for articles published between 1955 and 2024. The medications found were divided into classes and sub-classes, and the subclasses were graded according to their level of evidence. *Results*: From 12,097 results, 1115 were found to be DIM. The subclasses of medications with level A evidence were intravenous anesthetics (etomidate), cephalosporins (ceftazidime, cefepime), fluoroquinolones (ciprofloxacin), selective serotonin reuptake inhibitors (citalopram, escitalopram, paroxetine, sertraline), tricyclic antidepressant (amitriptyline), glutamate antagonist (amantadine), atypical antipsychotics (clozapine, quetiapine), antiseizure medications (carbamazepine, oxcarbazepine, phenytoin, gabapentin, pregabalin, valproate), pure opioid agonist (fentanyl, morphine), bismuth salts, and mood stabilizers (lithium). The single medication with the highest number of reports was etomidate. Drug-induced asterixis is associated with a specific list of medications. The neurotransmitters likely involved in DIM are serotonin, dopamine, gamma-aminobutyric acid (GABA), and glutamate. *Conclusions*: DIM may be reversible with management that can include drug discontinuation, dose adjustment, and the prescription of a medication used to treat idiopathic myoclonus. Based on the main clinical constellation of symptoms and pathophysiological mechanisms found in this study, DIM can be categorized into three types: type 1 (serotonin syndrome), type 2 (non-serotonin syndrome), and type 3 (unknown).

## 1. Introduction

Myoclonus is a hyperkinetic movement disorder characterized by sudden, brief, shock-like involuntary movements [1]. When there is a sudden muscular contraction, it is called a positive myoclonus; on the other hand, when a cessation of the ongoing muscular contraction occurs, it is called a negative myoclonus or asterixis [2]. The term “myoclonus” is derived from “paramyoklonus multiplex”, which was coined by Nikolaus Friedreich in a 50-year-old male patient while describing involuntary movements at rest [3].

Many classifications have been developed to categorize myoclonus. For example, myoclonus can be classified based on its source, such as cortical, subcortical, cortical–subcortical, subcortical–nonsegmental, and spinal (segmental, propriospinal, and peripheral) [4]. Also, it can be categorized based on its clinical distribution into focal, segmental, multifocal, and generalized [5]. The etiological classification of myoclonus is crucial because it can assist in clinical decisions regarding the pharmacological management of this movement disorder [6].

Myoclonus can be primary or secondary to infectious, metabolic, endocrine pathology, degenerative, inflammatory, toxic, genetic, and pharmacological causes [7]. Many examples of drug-induced myoclonus (DIM) are transient and occur in the setting of diffuse encephalopathy associated with drug toxicity, acute metabolic abnormalities, or infectious disorders [8]. In most cases, conducting a detailed investigation with neurophysiological studies is not practical.

The epidemiological data on DIM are scarce. A French pharmacovigilance database study revealed that the incidence of DIM is around 0.2% in the general population; the most frequent culprit drugs were antibiotics, antidepressants, anxiolytics, and opiate agents. Also, the authors reported that the median age of patients was 55 years, and 10% of these patients had a concomitant neurological disease [9]. Caviness et al. reported in Olmsted County, from 1976 to 1990, a lifetime prevalence of 8.6 cases per 100,000 people, while the annual incidence rate was 1.3 cases per 100,000 person-years [10]. Thwaites et al. showed no age and sex predilection for hydromorphone-induced myoclonus in terminally ill hospice patients [11]. Janssen et al. found that the most common classes of medications associated with myoclonus were opiates, antidepressants, antipsychotics, and antibiotics [5].

More research on this subject is needed because myoclonus can impact patients and caregivers, causing increased morbidity and a higher economic burden for the healthcare system. In this systematic review, we will discuss myoclonus secondary to drugs, providing a list of drugs along with the level of evidence of their association with myoclonus. We highlight the importance of considering myoclonus as a potential side effect of various drugs, even at low doses, to avoid costly and unnecessary investigations, as well as excessive treatments.

## 2. Methodology

We searched the Medline/PubMed database to locate all existing reports on myoclonus secondary to medications published from January 1955 to June 2024 in electronic form. The search term was “myoclonus”. The query was (“myoclonus”[MeSH Terms] OR “myoclonus”[All Fields]). For a complete list of all the publications concerning DIM in PubMed, consider reading Appendix A. The Preferred Reporting Items for Systematic reviews and Meta-Analyses (PRISMA) statement published in 2020 was used for the literature search and methodology [12].

The inclusion criteria covered case reports, case series, original articles, letters to the editor, bulletins, and poster presentations, published from January 1955 to June 2024, without language restriction, to ensure a thorough review. When the non-English literature was beyond the authors’ proficiency (English, French, German, and Spanish) or the English abstract did not provide enough data, such as articles in Danish, Japanese, and Polish, the Google Translate service was used [13].

The authors independently screened the titles and abstracts of all articles from the initial search. Disagreements between authors were solved through discussion. Cases not accessible by electronic methods, including after a formal request to the authors, were excluded. Cases with more than one factor contributing to the myoclonus were evaluated based on the probability of the event occurring, using the Naranjo algorithm.

From 12,097 results, 1115 were found to be DIM (Figure 1). Data abstraction was performed. The authors read the title and the abstract of all the articles found in the initial search. When provided, we extracted the PubMed identifier (PMID), the main cause of myoclonus, and the year of publication. The data were extracted by two independent authors and double-checked to ensure that they matched.

No statistical analysis was performed. The articles are ordered in Appendix A according to their PubMed identifier (PMID) number.

The clinical definition of myoclonus was obtained from Caviness et al. [6]. The Naranjo algorithm was used to determine the likelihood of whether an adverse drug reaction was actually due to the drug rather than the result of other factors [14]. After the literature review, a grading system was developed based on the quantity of patients with DIM and the quality of the published manuscripts. An “A” level of evidence was defined when there were more than 20 individuals reported to have myoclonus caused by that specific class of medications. “B” was characterized by 5 and 20 individuals reported with DIM. “C” was defined by less than five individuals reported to have myoclonus. When there was no case report and only animal studies, the subclass of drugs was graded as “D”.

## 3. Results

The classes of drugs were categorized as anesthetics, antibiotics, antidementia, antidepressants, antiemetics, antihemorrhagic, antihistamines, antineoplastic and immunosuppressive agents, antiparasitic, antiparkinsonian, antipsychotics, antiseizure, antiviral, anxiolytics, cardiovascular, opioids, and others. We also included a specific category regarding the drugs related to animal models of myoclonus. Table 1 is a summary of the classes of medications already associated with myoclonus and their level of evidence.

## 4. Anesthetic Agents

Enflurane [15], etomidate [16], nitrous oxide [17], and propofol [18] are some of the anesthetic agents associated with myoclonus. Deep anesthesia and muscle relaxants may attenuate myoclonus [19]. However, it is worth mentioning that an association between local, spinal anesthesia and myoclonus has already been reported [20]. For anesthetics associated with myoclonus in the literature, consider reading Appendix A [15,16,17,18,21,22,23,24,25,26,27,28,29,30].

In an experimental study, enflurane and isoflurane, compared to halothane, were more commonly associated with myoclonus in mild hypocapnic cats and provoked greater airway inflammation [31]. Ng et al. reported a case of enflurane-induced myoclonus involving multiple muscle groups, except those innervated by the cranial nerves [15]. Interestingly, nitrous oxide-induced myoclonus was already related to a propriospinal source due to subacute spinal cord degeneration associated with cobalamin deficiency [32]. A similar hypothesis can be assumed for cases of myoclonus induced by cobalamin supplementation [33].

Propofol is a short-acting gamma-aminobutyric acid receptor (GABA-A) agonist that can safely be used as a general anesthetic. Propofol is already associated with causing [18] and improving [34] myoclonus. Walder et al. found that propofol-induced seizure-like phenomena are most common during induction, emergence, or the immediate postoperative periods [35]. Chao et al. proposed that propofol-induced myoclonus occurs due to cortical reflex myoclonus [36].

Ketamine-induced myoclonus was rarely described in the literature. There are only studies with veterinary anesthesia in which dogs showed myoclonus [25]. However, no myoclonus was observed in humans, suggesting something specific to the metabolism of this medication in some dog breeds, especially the English Greyhound [37].

Etomidate was the most common medication reported to cause myoclonus in the literature. Several articles published were clinical trials assessing the possible management of etomidate-induced myoclonus. Doenicke et al. reported that the incidence and intensity of myoclonus after induction with etomidate are dose-related, suppressed by pre-treatment, and unassociated with seizure-like EEG activity [38]. Etomidate-induced myoclonus likely involves neocortical glutamate accumulation and N-methyl-D-aspartate receptor (NMDAR) modulation activity. Myoclonus was correlated with the NMDAR-induced downregulation of potassium-chloride transporter member 5 (KCC2) protein expression [39]. Therefore, suppressing the astrogliosis in the neocortex and promoting extracellular glutamate uptake by regulating glutamate transporters in the motor cortex may be a therapeutic option for managing myoclonus associated with etomidate [40].

Zhou et al. showed that the incidence of etomidate-induced myoclonus was significantly lower in midazolam-treated groups (RR = 0.34, 95% CI [0.26, 0.44], *p* < 0.05). The authors revealed that subgroups divided by the degree of myoclonus showed a significantly lower incidence of mild myoclonus (RR = 0.56, 95% CI [0.39, 0.80], *p* < 0.05), moderate myoclonus (RR = 0.20, 95% CI [0.10, 0.41], *p* < 0.05), and severe myoclonus (RR = 0.12, 95% CI [0.04, 0.39], *p* < 0.05) [41]. Nooraei et al. showed that using a priming dose of atracurium efficiently suppresses etomidate-induced myoclonus during the induction of anesthesia. The adjusted odds ratio in this model of myoclonus in the control group was 6.6 (95% Cl [1.5–9.7], *p* < 0.05) [42]. Collin et al. reported that alfentanil significantly reduced myoclonus associated with etomidate [43]. Du et al. found that etomidate-induced myoclonus in the dexmedetomidine-treated groups was significantly lower than that in the control groups (RR = 0.27, 95% CI [0.15, 0.47], *p* < 0.00001) [44]. Feng et al. reported that etomidate-induced myoclonus in the propofol-treated groups was significantly lower than that in the control groups (RR = 2.99, 95% CI [2.40, 3.71], *p* < 0.0001) [34]. Hua et al. reported that etomidate-induced myoclonus in the butorphanol-treated groups was significantly lower than that in the control groups (RR = 0.15, 95% CI [0.10, 0.22], *p* < 0.00001) [45]. Zhu et al. showed that the pre-injection of dezocine (opioid analgesic) can reduce the incidence of etomidate-induced myoclonus (RR = 0.25, 95% CI [0.13, 0.50], *p* < 0.0001) [46]. Finally, Greenwood et al. found an absolute reduction in risk with prophylactic medications, ranging from 47% to 81% for mild, 52% to 92% for moderate, and 61% to 96% for severe myoclonus. Also, the authors observed that opioids have a consistent and substantial effect on the reduction in myoclonus [47]. Further studies assessing all the therapeutic options already proposed for etomidate-induced myoclonus are mandatory. Studying this specific cause of myoclonus can be used to understand DIM pathophysiology and further improve current management strategies.

Spinal myoclonus following neuraxial anesthesia was rarely reported in the literature. Shiratori et al. found 23 cases of spinal myoclonus associated with local anesthetics. In total, 82.6% of the cases occurred following lumbar spinal anesthesia, and the rest following epidural anesthesia. Amide-type local anesthetics, such as dibucaine, lidocaine, bupivacaine, prilocaine, and levobupivacaine, were used in 95.7% of the cases, whereas an ester-type local anesthetic, such as tetracaine, was used in only one case [48].

## 5. Antibiotics

Stimulus-sensitive myoclonus and encephalopathy have already been associated with cephalosporins [49], carbenicillin [50], imipenem [51], quinolones [52], penicillin [53], piperacillin [54], and ticarcillin [55]. For antibiotic medications associated with myoclonus in the literature, consider reading Appendix A [50,54,56,57,58,59,60,61,62,63,64,65,66,67,68,69,70,71,72,73,74,75,76,77,78,79,80,81,82,83,84,85,86].

### 5.1. Penicillins, Cephalosporins

Antibiotics like penicillin and cephalosporin have been shown to cause myoclonic jerks, which can be generalized, multifocal, or segmental [87]. Myoclonus is commonly accompanied by other symptoms, such as an altered mental state, seizures, aphasia, chorea, and a skin rash [5]. Bhattacharyya et al. systematically reviewed encephalopathy secondary to antibiotics and found three clinical phenotypes. Type 1 was characterized by onset within days of antibiotic initiation, the occurrence of myoclonus or seizures, and resolution within days. Type 2 was marked by onset within days of antibiotic initiation, the frequent occurrence of psychosis, and the rare occurrence of seizures. Type 3, seen only with metronidazole, is characterized by the onset of encephalopathy within weeks after the initiation of metronidazole, the frequent occurrence of cerebellar dysfunction, and rare seizures [88].

Conway et al. reported five cases of endocarditis in which high doses of penicillin resulted in neurotoxicity in the form of drowsiness and myoclonus. The adverse effect was unrelated to the type of penicillin preparation (sodium or potassium) used [89]. Interestingly, other penicillin formulations or penicillin itself provoked the same neurotoxicity manifestations [90]. Also, Lerner et al. found higher concentrations of penicillin in the cerebrospinal fluid (CSF), which they attributed to the altered blood–brain barrier (BBB) permeability caused by uremia and the deterioration of renal function, increasing the toxic levels [90]. Penicillins are excreted by the kidneys into the urine in patients with normal renal function. Hence, the penicillin dose must be adjusted according to the renal function of the patient to prevent side effects [91]. Of note, some authors reported the occurrence of myoclonus with penicillins and the association between cephalosporins and renal clearance impairment [92], but this should be evaluated with caution because end-stage renal disease is also associated with myoclonus [93].

The cephalosporins most commonly associated with myoclonus were ceftazidime and cefepime. One of the mechanisms causing cephalosporin neurotoxicity is the induction of endotoxins and, possibly, glutaminergic mechanisms. Laboratory studies also show that cephalosporins have a high affinity for GABA-A receptors, which cause high penetrance through the BBB and are more neurotoxic [94]. Studies show that in patients with renal disease, the maintenance dose should be reduced, and the patients should be monitored for neurotoxicity. Care should be taken when using cefepime use and its toxicity should be kept in mind whenever a patient receiving it shows a change in mental status or myoclonus [95]. Chow et al. retrospectively studied 42 cases of cefepime and 12 cases of ceftazidime-induced neurotoxicity with myoclonus. In total, 33% of the individuals were already uremic, which led to a delay in diagnosis due to this confounding factor. The median time between the onset of symptoms and diagnosis for cefepime-induced encephalopathy compared to ceftazidime-induced neurotoxicity was 5 and 3 days, respectively [*p* = 0.005] [96].

### 5.2. Fluoroquinolones and Quinolones

Myoclonus is commonly associated with fluoroquinolones, especially ciprofloxacin. Some authors have called this association “ciproclonus” [74]. Ciprofloxacin is also associated with propriospinal myoclonus by antagonizing gamma-aminobutyric acid metabolism [73]. Another case reported that myoclonus is associated with delirium, and the authors highlighted the importance of ciprofloxacin as a cause of delirium and myoclonus in elderly patients [72]. Also, consideration should be given to prescribing reduced-dose ciprofloxacin to elderly patients with renal impairment. Rissardo et al. reported that the fluoroquinolone-induced myoclonus distribution is focal, multifocal, segmental, axial, and generalized. Myoclonus was the most common movement disorder associated with fluoroquinolones, and the authors found 25 cases in the literature. Finally, the authors described that previously reported fluoroquinolone-associated myoclonus was likely related to an increased glutamate concentration due to the neurotoxic effects of fluoroquinolones, including oxidative process, chelated cations, and disturbed gene expression [97].

### 5.3. Other Classes of Antibiotics

Carbapenems are commonly associated with seizures, which can also explain the high number of reports of myoclonus. This can suggest a cortical source for the myoclonus secondary to carbapenems. Cannon et al. systematically reviewed the literature regarding carbapenems and their risk of seizure, and the authors found that imipenem was more epileptogenic than non-carbapenem antibiotics. But there was no statistically significant difference between the imipenem and the meropenem [98]. Noteworthy, after cephalosporins and fluoroquinolones, carbapenem and penicillins were the most common classes of antibiotic-induced myoclonus.

Macrolides such as erythromycin and azithromycin were rarely reported to be associated with myoclonus. It is worth mentioning that there are some rare cases of azithromycin-induced myoclonus. Still, these cases can also be explained by encephalitis lethargica, commonly associated with some viral and bacterial pathogens [80].

Linezolid-induced myoclonus was eventually described in the literature [86]. The role of linezolid as a weak, non-selective, reversible monoamine oxidase inhibitor can explain the reasonable number of articles about myoclonus. There have been post-marketing reports of serotonin syndrome when linezolid was given with or soon after the discontinuation of serotonergic drugs [99].

Gentamicin-induced multifocal myoclonus was observed in elderly individuals with renal impairment, and a favorable prognosis on gentamicin discontinuation was noticed [81]. Also, myoclonus was reported with anthelmintics and antituberculosis drugs such as piperazine [100] and isoniazid [101], respectively. Other classes of antibiotics were rarely associated with myoclonus, such as lipopeptides [83], glycopeptides [84], and tetracyclines [85].

## 6. Antidementia

Myoclonus secondary to memantine was observed in patients with dementia [102]. Pei et al. found five cases in the literature of memantine-induced myoclonus. The authors reported that the onset of myoclonus after memantine exposure ranged from 6 days to 2 months, with the complete resolution of myoclonus upon the cessation of memantine [103]. The mechanism underlying memantine-induced myoclonus remains unclear but might involve altered dopamine, serotonin, and glutamate levels. Noteworthy, memantine is an amino-adamantane chemically similar to amantadine [104].

Rissardo et al. reported a case of action myoclonus secondary to donepezil. The authors also found six other reports of myoclonus secondary to donepezil/galantamine, but no report of rivastigmine-induced myoclonus was identified. Also, they observed that the most frequent presentation was multifocal myoclonus [105]. In experimental studies, reducing acetylcholine function improved picrotoxin-induced myoclonus [106]. Therefore, high levels of acetylcholine may be associated with the frequency of involuntary twitching. Noteworthy, myoclonus is common in Alzheimer’s disease and vascular dementia and can also be present in Lewy body dementia, although typically in the later stages of the disease [107].

## 7. Antidepressants

During the last decade, 13.2% of adults have used at least one antidepressant in the past 30 days, which is higher among females (17.7%) than males (8.4%) [108]. Also, the prescription of antidepressants, especially SSRIs and SNRIs, has increased over the last few years due to the COVID-19 pandemic [109]. In 2006, The US Food and Drug Administration (FDA) issued an advisory about the risk of serotonin syndrome (which has myoclonus as a component) associated with the concomitant use of drugs from two widely prescribed medication classes of this family, namely selective serotonin reuptake inhibitor (SSRI) and selective norepinephrine reuptake inhibitor (SNRI). Interestingly, among the antidepressants, SSRIs [110] and tricyclic antidepressants (TCA) [111] were the most common classes associated with myoclonus. For antidepressant medications associated with myoclonus in the literature, consider reading Appendix A [111,112,113,114,115,116,117,118,119,120,121,122,123,124,125,126,127,128,129,130,131,132,133,134,135].

A 62-year-old woman with a history of congestive heart failure and coronary artery disease, mitral valve stenosis, and chronic kidney disease received buspirone and, within a day, developed dramatic myoclonus, which did not improve with intramuscular diphenhydramine. However, the myoclonus completely improved after clonazepam administration [136]. Riaz et al. found only eight cases of bupropion-induced myoclonus, in which the most common association with myoclonus in younger individuals was overdose and intoxication. Bupropion has higher rates of seizures compared to other antidepressants, which can occur even in the presence of benzodiazepines. The authors hypothesized that the cause of seizures could be related to the antagonization of nicotinic acetylcholine receptors [122]. Also, there are reports of myoclonus related to monoamine oxidase inhibitors [137] and trazodone [126].

Janssen et al. reported that the distribution of myoclonus associated with serotonin reuptake inhibitors is mainly multifocal and generalized [5]. On the other hand, TCAs are related to focal (especially jaw), multifocal, and generalized myoclonus. Garvey et al. reported a prevalence of 30% of myoclonus in individuals using TCA, but only 9% was clinically significant [138]. Rissardo et al. found 26 cases of myoclonus associated with amitriptyline. They correlated the occurrence of myoclonus with the effect of amitriptyline on the serotonin receptors, and a dose-dependent effect was observed [111].

The mechanism of antidepressant-induced myoclonus is unclear but may be related to increased serotonergic transmission. A study showed EEG and evoked potential abnormalities in TCA-induced myoclonus [92]. In another post-marketing pharmacovigilance database study, phenelzine was associated with the highest reported adjusted odds ratio for antidepressant-induced movement disorders, followed by clomipramine [139]. SSRIs increase serotonin levels in the synaptic cleft, and TCAs increase serotonin activity. Interestingly, a combination of TCA and lithium appears to have a compound effect more likely to cause myoclonus than these drugs administered in an isolated fashion [140].

## 8. Antiemetics

Dopamine and 5HT3 antagonists have already been associated with myoclonus. Due to their dopamine receptor blocker properties, most antiemetics, such as prochlorperazine, promethazine, and metoclopramide, may be associated with some tardive disorders [141].

Metoclopramide was commonly associated with myoclonus, but this medication alone was never related to myoclonus [142]. Metoclopramide is believed to have a 5-HT3 receptor-blocking effect; although this effect is weak, it may influence the development of myoclonus [143]. Hyser et al. reported a case of metoclopramide-induced multifocal myoclonic jerking in the setting of chronic kidney disease, which ceased after the discontinuation of metoclopramide [144]. Harada et al. reported a 40-year-old patient who was observed to be unresponsive, with occasional myoclonus in the legs following an intramuscular injection of metoclopramide [143].

The dopaminergic, cholinergic, and GABA-ergic systems may be involved in promethazine-induced movement disorders [145]. Dy et al. reported a 63-year-old patient experiencing inducible myoclonus after receiving dextromethorphan–promethazine cough syrup for an upper respiratory infection [146].

Other antiemetics associated with myoclonus were ondansetron [147] and palonosetron [148]. Interestingly, palonosetron has allosteric interactions and positive cooperativity with 5-HT3 receptors, while ondansetron has simple bimolecular competitive binding. Also, palonosetron may trigger 5-HT3 receptor internalization and the degradation of the internalized receptor, which reduces receptor density at the cell surface and leads to the prolonged inhibition of 5-HT3 receptor function [149].

## 9. Antihistamines

Cimetidine-induced myoclonus is usually associated with dose-dependent encephalopathy and is more frequently observed in renal and hepatic dysfunction in elderly individuals. Other symptoms, besides myoclonus, include mild disorientation and psychosis; the symptoms fully improve after the discontinuation of cimetidine [150]. Oxatomide, a second-generation antihistamine, is prescribed for allergies in Europe and Japan, and has already been reported with myoclonus. The authors hypothesized that minor toxic–metabolic disturbance caused by antihistamine use might have driven their elderly subject, a myoclonus-prone patient, into a transient myoclonic state [151]. Other antihistamines associated with myoclonus were triprolidine [152] and tripelennamine [153].

## 10. Antineoplastic

Among the antineoplastic and immunosuppressive agents, alkylating agents were the most common class associated with myoclonus. Chlorambucil-induced myoclonus was associated with therapeutic doses and intoxication levels [154]. The electroencephalographic findings were generalized slowing or paroxysms of high-amplitude spike–wave activity [155]. The discontinuation of this nitrogen mustard agent usually improves the myoclonus [156]. Other alkylating agents associated with myoclonus were cyclophosphamide [157], ifosfamide [158], and busulfan [159].

Monoclonal antibodies like ipilimumab [160], nivolumab [161], and pembrolizumab [162] have already been reported to be associated with myoclonus. Ipilimumab, a fully human IgG1 monoclonal antibody against cytotoxic T-lymphocyte-associated protein 4 (CTLA-4), is an effective treatment for melanoma. Nivolumab, a fully human IgG4 monoclonal antibody against programmed cell death protein 1 (PD-1), is an effective treatment for non-small cell lung cancer and melanoma, among other cancers [163]. Also, the nitrogen mustard derivative prednimustine [164] and some nucleoside analogs, such as 5-fluorouracil [165], floxuridine [166], and pentostatin [167], were associated with myoclonus.

Tacrolimus-induced severe neurotoxicity in the form of myoclonus, seizures, and leukoencephalopathy is uncommon and occasionally reported in kidney transplant recipients. Tacrolimus has been reported to cause generalized myoclonus, myoclonus of both lower limbs, and segmental myoclonus of the abdominal wall [168]. Cyclosporine is a lipophilic, cyclic oligopeptide that inhibits calcineurin and modulates the immune system by suppressing T-cell proliferation. Tremor is the most common neurological complication of cyclosporine [169]. Kang et al. reported a case of opsoclonus–myoclonus secondary to cyclosporine therapy. The authors believe that the highly lipophilic nature of cyclosporine allows it to cross the BBB and provoke changes in neurotransmission through altered dopamine receptor function [170].

Topoisomerase drugs were rarely associated with myoclonus. There are isolated reports associating myoclonus with irinotecan and pyrazoloacridine [171]. Shun et al. suggested that benzodiazepine should be considered the drug of choice in managing irinotecan-induced myoclonus [172].

## 11. Antiparkinsonian

The antiparkinsonian medications most commonly associated with myoclonus were levodopa and amantadine. Rissardo et al. found 22 individuals with amantadine-associated myoclonus reported in the literature [173]. The exact mechanism causing this side effect is unknown. However, it is thought that it could act by increasing stimulatory neurotransmitters like dopamine, norepinephrine, and sigma-1 receptors in the cortico-striato-pallido-thalamo-cortical loop in the brain and antagonizing inhibitory neurotransmitters like acetylcholine and glutamate [174]. Interestingly, amantadine has been shown to cause craniofacial myoclonus [175].

Dopamine precursors, especially levodopa, were among the most common classes of medication associated with myoclonus. In 1969, Cotzias et al. reported the first cases of myoclonus associated with levodopa [176]. In 1975, Klawans et al. described 12 individuals who developed levodopa-induced myoclonus after at least one year of levodopa. They described that the myoclonus was bilateral and was not interrupted by the sleep cycle; also, seven individuals developed choreiform dyskinesias [177]. Nausieda et al. found a correlation between the prevalence and severity of myoclonus and the duration of levodopa therapy [178]. Luquin et al. reported an incidence of 3.5% of myoclonus in individuals with PD with levodopa therapy [179]. Also, apparently, PD progression influences the incidence of myoclonus. Marconi et al. revealed an incidence of 66% in individuals with at least ten years of levodopa therapy [180]. Cases of negative myoclonus and seizures were also associated with levodopa therapy [181]. Interestingly, in old literature, myoclonus secondary to levodopa was included in the spectrum of levodopa-induced dyskinesia [182], which can be a confusing term due to the current understanding of myoclonus pathophysiology with electrodiagnostic studies.

Vardi et al. reported six cases of myoclonus associated with bromocriptine [183]. Tandberg et al. noticed that individuals recently diagnosed with PD without therapy have an increased incidence of myoclonus compared to healthy individuals [184]. Other antiparkinsonian drugs associated with myoclonus were trihexyphenidyl [185], entacapone [186], pramipexole [187], and selegiline [9].

## 12. Antipsychotics

There are reports of myoclonus associated with typical and atypical antipsychotics. Historically, typical antipsychotics were more frequently reported with myoclonus, but recent studies showed that atypical antipsychotics are more commonly associated. Haddad et al. proposed that atypical antipsychotics lower the seizure threshold to a more significant degree than typical antipsychotics [188]. The incidence of clinical seizures with antipsychotics is 0.5–1.2% in patients without a history of epilepsy, and only EEG seizures are observed in around 7% of the individuals [189]. However, the low rate of recognition may be due to the complexity of psychiatric illness symptoms and the atypicality of psychomotor seizures. Also, the number of prescriptions for atypical compared to typical antipsychotics has significantly increased in the last few decades [190]. Janssen et al. reported that the myoclonus distribution was multifocal for typical and multifocal to generalized for atypical [5]. For antipsychotic medications associated with myoclonus in the literature, consider reading Appendix A [191,192,193,194,195,196,197,198,199,200].

Chlorpromazine and clozapine, both antipsychotics, have a relatively high potential to induce seizures [201]. Clozapine is already associated with all types of seizures with and without impaired awareness. Although generalized tonic–clonic seizures are frequently observed, myoclonic seizures may occur and may be encountered in daily practice [202]. Also, among the antipsychotics, clozapine stands out for its heightened seizure risks, especially during titration and at high doses, necessitating close monitoring and individualized approaches [203].

Altıparmak studied antipsychotic-induced myoclonus in an inpatient psychiatric hospital. Six of the ten patients in the study who developed myoclonus received clozapine, two received olanzapine, one received amisulpride, and one received quetiapine. The mean age was 24.2 years, and the mean duration of the psychiatric disorder was 33.5 months. Valproic acid was prescribed to eight individuals to control these myoclonic seizures, while lorazepam and clonazepam were used in others [192]. Of note, levetiracetam can also be used for the management of myoclonus but is not commonly prescribed in clinical practice in psychiatric wards due to dose-independent aggression and psychotic side effects.

Tominaga et al. proposed the term “tardive myoclonus”, which was defined as a postural myoclonus associated with the long-term use of antipsychotic therapy [204]. Fukuzako et al. reported that 38% of the individuals in a psychiatry hospital had tardive myoclonus; they also observed that the antipsychotic dose was significantly higher in these individuals compared to those who did not develop tardive myoclonus [205]. Ortí-Pareja et al. reported an incidence of only 1% of tardive myoclonus, but the population assessed was those referred to the neurology service [206]. Interestingly, Little et al. reported a case of myoclonus after five months of antipsychotic withdrawal [207]. Also, Staedt et al. described nocturnal myoclonus in all individuals with schizophrenia and long-term antipsychotic therapy [208].

Myoclonus was observed with subtherapeutic, therapeutic, and higher doses of quetiapine. Uvais et al. reported a case of quetiapine-induced myoclonus that was sensitive to posture (more in lying down position) in a 64-year-old male diagnosed with mild depression and insomnia, even with a single low dose of 12.5 mg per night [209]. Aggarwal et al. presented two cases of probable quetiapine-induced myoclonus at high doses (400–800 mg). The first individual was a 19-year-old female with mania who was started on 400 mg/day quetiapine and developed right upper limb myoclonic jerks, which were resolved by reducing the dose to 200 mg/day. The second case was a 17-year-old female with schizophrenia who developed myoclonus on clozapine 250 mg and then, after an adequate washout period, again developed myoclonus while on 600 mg/day quetiapine. Both cases fully recovered after reducing the dose of quetiapine [197]. In addition, Baysal Kirac et al. reported a case of a patient with dementia and a positive family history of juvenile myoclonic epilepsy who was given quetiapine and then developed myoclonic status epilepticus within one month [210]. Velayudhan et al. also reported a 64-year-old man with schizophrenia who developed myoclonus after quetiapine 800 mg/day was introduced; the myoclonus resolved after reducing the dose to 400 mg/day [211].

The pathophysiology of antipsychotic-induced myoclonus is not fully understood. However, the action of quetiapine on serotonergic, dopaminergic, and gamma-aminobutyric acid (GABA)-ergic mechanisms can potentially cause myoclonic jerks [5].

## 13. Antiseizure Medications

Magaudda et al. reported a 31-year-old male with idiosyncratic epilepsy taking carbamazepine (CBZ) 800 mg/day, who developed a subcortical myoclonus involving the right thumb and shoulder [212]. The myoclonic jerks improved with carbamazepine withdrawal, and they returned with the carbamazepine rechallenge. Similar observations were already described in adults and pediatric individuals. Interestingly, the plasma carbamazepine levels and EEG were normal [213]. Dhuna et al. presented a child who developed subcortical, multifocal myoclonus, which resolved within 24 h after carbamazepine was discontinued. This exacerbation occurred with therapeutic carbamazepine serum levels and was thought to be related to the toxic levels of carbamazepine-10,11-epoxide (CBZE) metabolite [214]. Holtmann et al. stated that carbamazepine-induced myoclonus is idiosyncratic rather than dose-related [215]. Parmeggiani et al. proposed that increased cortical inhibition could be the electrophysiological basis of carbamazepine-induced asterixis. They also proposed that the presence of spike–wave (rather than sharp wave) discharges in children with benign epilepsy with centro-temporal spikes (BECTS) might be used as an electrophysiological predictor of an abnormal response to CBZ [216]. For antiseizure medications associated with myoclonus in the literature, consider reading Appendix A [217,218,219,220,221,222,223,224,225,226,227,228,229,230,231,232].

## 14. Antiviral

Ganciclovir, acyclovir, valacyclovir, and foscarnet are all known to cause neurotoxic side effects, albeit rarely. Use may lead to tremors, myoclonus, dysarthria, ataxia, delirium, hallucinations, and lethargy [233]. Acyclovir-induced myoclonus was uncommonly reported in the literature, but the first trials of this medication revealed that myoclonus occurred in 18% of the individuals, likely associated with intoxication [234]. Haefeli et al. found that acyclovir caused tremor/myoclonus in more than half of the subjects [235]. In other studies, a rare reversible encephalopathy occurring in less than one percent of the individuals treated with conventional doses of acyclovir was reported [236]. Other symptoms of acyclovir encephalopathy were disorientation (58%), decreased consciousness (38%), hallucinations (36%), agitation (27%), and dysarthria (19%) [234].

Vidarabine, another antiviral related to purines, was also associated with myoclonus [237]. However, no antiviral medication pyrimidine analog was found to be associated with myoclonus in the literature, including animal studies.

## 15. Anxiolytics

Myoclonus was associated with the initiation of some anxiolytic drugs and also with their withdrawal; some examples of this association are buspirone [238], carisoprodol [239], lorazepam [240], and midazolam [23].

Carisoprodol-induced myoclonus was associated with overdose in the literature. In 2012, carisoprodol was placed on Schedule IV by the Drug Enforcement Administration [241]. Carisoprodol is not detected on all toxicology tests, which may delay the diagnosis of overdose. A cortical myoclonus source was observed in individuals taking carisoprodol [239].

In experimental studies, the Guinea baboon (Papio papio) can present two different types of myoclonus. One type, induced by photic stimulation (intermittent luminous stimulation) preceded by paroxysmal discharges, can be blocked by benzodiazepines. On the other hand, type two of myoclonus may be facilitated by lorazepam and diazepam, lowering the seizure threshold observed on EEG [242]. Based on these observations, we can assume that the origin of benzodiazepine-induced myoclonus is related to a cortical source, which was interestingly noticed with other anxiolytics [243].

## 16. Cardiovascular

There are some isolated case reports of calcium channel blockers associated with myoclonus. Nifedipine was associated with myoclonus and dysarthria [244]. Verapamil was associated with myoclonic dystonia [245] and multifocal myoclonus [246]. Amlodipine-induced myoclonus was reported in the setting of stable chronic renal failure [247]. Diltiazem, in therapeutic doses and in combination with citalopram, was considered responsible for myoclonus while recumbent and in response to startling [248].

Carvedilol, a nonselective beta-adrenergic blocker, was reported to cause multifocal myoclonus without other clinical signs [249]. Other antihypertensives, such as ketanserin [250] and furosemide [9], were also associated with myoclonus. Interestingly, González et al. reported that enalapril-induced myoclonus was a dose-dependent side effect [251].

The fixed-dose combination of sacubitril/valsartan, usually prescribed for congestive heart failure, was associated with myoclonus [252]. In repeated dose studies in mice and rats but not in primates, increased locomotor activity, twitches, and sensitivity to touch were observed with sacubitril/valsartan [253]. Also, enkephalinase inhibitors, with neutral endopeptidase inhibitors being part of this family, were previously shown to interfere with the dopaminergic system in experimental models [254].

Vasopressor-induced myoclonus was already observed with dobutamine and midodrine. Sympathomimetics can facilitate neuromuscular transmission followed by the prolonged rapid stimulation of motor nerves, which is most likely mediated by alpha-1 receptors [255]. The cause of dobutamine-induced myoclonus is not yet fully understood, but it may be related to kidney failure and neurotoxicity [256]. It is suggested that inhibiting P-glycoprotein reduces the breakdown of dobutamine and makes it easier for the substance to enter the central nervous system. Also, chronic kidney disease may affect the half-life of dobutamine [257].

## 17. Opioids

Myoclonus may occur as a result of the initial administration [258], change [259], or withdrawal [260] of opiates. Janssen et al. reported that the distribution of opioid-induced myoclonus is usually generalized or multifocal, but focal cases were already described [5]. Also, myoclonus associated with opioids is frequently associated with the use of other medications, such as antidepressants and antipsychotics [261]. For opioids associated with myoclonus in the literature, consider reading Appendix A [9,261,262,263,264,265,266,267,268,269,270,271,272,273].

The long-term use of high doses of opiates, like in palliative care, was already correlated with an increased frequency of myoclonus. A retrospective review of 48 terminally ill hospice patients who received continuous parenteral hydromorphone for pain control was studied. The authors reported that agitation, myoclonus, and seizures were independently associated with the maximal dose (*p* < 0.05) and with the duration (*p* < 0.05) of continuous parenteral hydromorphone. A possible explanation for these findings could be the hydromorphone-3-glucuronide, a metabolic product of hydromorphone, which has been implicated in neuroexcitatory symptoms in laboratory investigations [11]. However, McCann et al. found no association between the plasma levels of morphine-3-glucuronide or hydromorphone-3-glucuronide and myoclonus [274].

Almedallah et al. reported a 24-year-old pregnant woman who was post-dated and had to undergo a cesarean section, for which epidural anesthesia with fentanyl, bupivacaine, and lidocaine was administered. She developed myoclonus, involving the upper and lower limbs, head, and torso while preserving consciousness. Myoclonus improved after fentanyl discontinuation, and no plasma concentration of fentanyl was noticed. She received morphine, tramadol, and lornoxicam without the occurrence of abnormal movements [24]. Similarly, Bruera et al. proposed the subcutaneous administration of an acute fentanyl overdose with 5000 mcg to a 62-year-old gentleman for the management of cancer pain, which resulted in generalized myoclonus along with confusion, restlessness, visual hallucinations, hyperalgesia, and tremors on the tactile stimulation of the arms and legs [275]. The pathophysiology proposed by some authors is general central excitability, with few cases reporting a full improvement of myoclonus after naloxone injections [276].

The management of morphine-induced myoclonus has already been reported in several cases in which the frequency of myoclonus was reduced with the prescription of clonazepam [277], dantrolene [278], and midazolam [279]. In experimental studies, ketamine improved morphine-induced hindlimb myoclonic seizures [280].

Scott et al. attributed this abnormal motor activity either to myoclonus produced by fentanyl causing the depression of higher central nervous system inhibitory centers or to extreme narcotic-induced rigidity [281]. Lane et al. found that fentanyl infusion withdrawal can cause myoclonus in the pediatric population [282]. A retrospective analysis described by Smith et al. showed that out of 127 surgical patients, 93 developed mild to moderate rigidity and received fentanyl, sufentanyl, and elfentanyl, as recorded by EEG and EMG [283]. It is noteworthy that some other anesthetics besides fentanyl can also cause myoclonus, so cases of fentanyl-induced myoclonus in surgical settings should be described with all the medications prescribed for the subject. Furthermore, a significant increase in myoclonus related to opioids, especially fentanyl, is likely to be observed in the following decades due to the “opioid crisis” [284].

## 18. Others

### 18.1. Heavy Metals

Bismuth salts are among the most commonly reported medications associated with myoclonus. They can cause encephalopathy and myoclonus, characterized by generalized, asymmetric, and stimulus-sensitive jerks during rest or action [285]. In severe cases, status epilepticus, coma, and death have been reported [286]. After drug withdrawal, the encephalopathy usually resolves, but approximately 10% of the individuals will have a bad outcome, including mortality and neuropsychiatric sequels [287]. Electroencephalographic findings are usually unspecific, and CSF analysis can show increased 5-hydroxy-indoleacetic acid levels [288].

### 18.2. Lithium

Rissardo et al. found 97 reports of lithium-induced myoclonus, and the distribution was focal, multifocal, and generalized. The mean age was 53.1, and the lithium dose was 942.7 mg/day [289]. Dyson et al. found an incidence of 25% of myoclonus in intoxicated individuals [290], and Bender et al. found a prevalence of 4.5% in individuals with affective disorder [291]. Based on the electrodiagnostic studies, we can hypothesize that myoclonus could occur due to dysfunctional cerebellar output, leading to cortical hyperexcitability [292]. Moreover, giant somatosensory evoked potentials are commonly reported as a remarkable feature of myoclonus occurring with lithium monotherapy or when combined with antidepressants [293].

## 19. Discussion

### 19.1. Drug-Induced Myoclonus Pathophysiological Mechanism

The myoclonus caused by most medications cannot be hypothesized because the reports did not include electrodiagnostic studies. Table 2 describes some of the proposed mechanisms for DIM. Increased serotonergic transmission may be the most commonly proposed pathophysiological mechanism for developing myoclonus [92]. Also, apparently, there are some risk factors associated with DIM, including advanced age, neurodegenerative comorbidities, a history of epilepsy, impaired renal function, electrolyte imbalance, and polypharmacy [9].

The neurotransmitters likely associated with DIM are serotonin, dopamine, GABA, and glutamate at various levels of the neuraxis (Figure 2). The schema of anatomical and neurophysiological mechanisms can help in clinical decisions and facilitate mechanism-based intervention. For anatomical localization, clinically, spinal segmental myoclonus can involve multiple groups of muscles innervated by 1–3 adjacent spinal levels, contrasting with peripherally generated myoclonus involving specific nerve roots, plexus, and peripheral nerves. Another interesting fact distinguishing propriospinal myoclonus from corticospinal myoclonus is the relatively slow conduction of propriospinal pathways related to high latencies involving the axial musculature compared to corticospinal origin. Nonetheless, adding to this, myoclonus with underlying encephalopathy can present with appendicular and axial myoclonus and cranial muscle involvement [308].

### 19.2. Drug-Induced Asterixis

Asterixis was uncommonly associated with medications (Appendix A). Asterixis can be attributed to the direct or indirect effect of the medications on the CNS. The direct effect can be explained by abnormalities in the neurotransmitter levels and toxic levels of the medications. On the other hand, the indirect effect is usually observed with medications that lead to hepatic dysfunction and increased levels of ammonia, predisposing the individual to the development of asterixis [181]. Drug-induced asterixis is restricted to a lower number of medications compared to myoclonus in general, so this can help in the differential diagnosis of patients presenting with asterixis. For medications associated with asterixis in the literature, consider reading Appendix A [58,142,145,147,148,173,309,310,311,312,313,314,315,316,317,318,319,320,321,322,323,324,325,326,327,328,329,330].

### 19.3. Proposed Classifications

We would like to propose a classification for DIM based on its symptomatology and main mechanism (Table 3). This classification is not strict; a significant overlap can be observed with a single medication. For example, valproate can be associated with types 2A and 2B, but it is not usually related to type 3. We noticed these features after observing some common characteristics in the literature and after reports of some rare and anecdotal drugs causing myoclonus.

First, the most commonly reported etiology for DIM is probably serotonin syndrome. Some patients developed the full syndrome, and others with the same combination of medications developed only myoclonus. The isolated myoclonus is part of a spectrum, which we would like to call the serotonin syndrome spectrum. The most common criteria used for diagnosing serotonin syndrome are Sternbach’s, Radomski’s, and Hunter’s criteria [331].

Second, some patients developed clinical encephalopathy before the development of myoclonus, without the fulfillment of serotonin syndrome criteria. In this context, we found two groups of individuals: one that developed associated liver injury and the other that did not have liver injury. Both groups of individuals had an overall worse prognosis compared to serotonin syndrome alone. It is noteworthy that the non-hepatic encephalopathy associated with myoclonus can also be categorized as drug-induced Creutzfeldt–Jakob-like syndrome. Drug-induced Creutzfeldt–Jakob-like syndrome is characterized by confusion, myoclonus, and EEG abnormalities associated with the use of a drug [332]. The most frequently reported medications associated with this specific constellation of symptoms are lithium [289], TCA [111], and carbamazepine [218].

A third group of individuals were observed to have only myoclonus and did not develop any other sign of serotonin syndrome or encephalopathy. Interestingly, this group of patients had an overall better prognosis than the first and second groups. However, further investigations regarding the management and follow-up of these individuals are needed.

### 19.4. Management

There has been an increased number of medications associated with myoclonus within recent decades. The recommended approach is to follow systematic reviews that have already been performed with the specific medication class, but the number of these studies is still limited. Therefore, we would like to provide some basic rules for managing DIM.

Management usually includes stopping the offending drug or modifying its dosage. However, this is a complex recommendation in clinical practice since sometimes it is not feasible due to underlying neuropsychiatric conditions, along with the risks and benefits of the responsible drug. Other options include adding specific treatments like drugs that act on the idiopathic myoclonus, such as benzodiazepines and antiseizure medications [92]. Another possible approach is noninvasive brain stimulation procedures, such as transcranial magnetic and electrical stimulation techniques. Occupational therapy, speech therapy, and physiotherapy can also be implemented in selected and resistant cases [333].

Classifying myoclonus on neurophysiological subtypes is important for management. Pena et al. explained that levetiracetam, valproic acid, and clonazepam are often used to treat cortical myoclonus. For cortical–subcortical myoclonus, the treatment of myoclonus is prioritized; hence, valproic acid is the mainstay of therapy. Subcortical–nonsegmental myoclonus can be treated with clonazepam. However, many other drugs can be used, according to the etiology. Segmental and peripheral myoclonus are difficult to treat and often remain resistant to treatment, but anticonvulsants and botulinum toxin injections can be attempted. The choice of drugs depends on the adverse effects, efficacy, and evidence-based knowledge, which is limited and unreliable due to nonstandard data [7]. Noteworthy, this approach is based on idiopathic myoclonus, so similar results may not be obtained with DIM (Figure 3).

## 20. Future Studies

Significant progress has been made in understanding the pathophysiology and management of idiopathic myoclonus. However, the literature on drug-induced myoclonus is scarce. Future studies should assess the epidemiological data regarding myoclonus secondary to medications. Online databases such as the FDA Adverse Event Reporting System (FAERS) should register subjects with more clinical details, including basic demographic features and electrodiagnostic studies. Most data published about myoclonus still lack basic descriptions of EEG and EMG, and therefore no source can be identified, which can result in misleading information about management.

The effect of medications on different neurotransmitters should be further investigated. There are studies in the literature with completely opposite results regarding neurotransmitters when using the same medication. Also, it is not uncommon to find published studies in the literature that were never replicated. In this way, there could be misleading information about attempts to provide pathophysiological explanations based on neurotransmitters.

While an increase in cases of DIM has been observed, more epidemiological data are required to understand the drugs’ nature, side effects, management, economic burden, and prognosis. More studies need to be performed on stable populations, with more pharmacodynamic investigations of the pharmacological agents. Similarly, more studies should be conducted by combining the medications used to control myoclonic symptoms. Ideally, to maximize reliability, multicenter clinical trials must be performed on cohorts of patients with myoclonus but without underlying neurological conditions and renal dysfunction. The lack of homogenous clinical features means that the number of drugs that could possibly cause myoclonus is underestimated. Providing reliable data about the prevalence and outcomes of this motor side effect to patients and caregivers when using individual medications in various healthcare settings remains a challenge.

## 21. Conclusions

The subclasses of medications with level A evidence were intravenous anesthetics (etomidate), cephalosporins (ceftazidime, cefepime), fluoroquinolones (ciprofloxacin), selective serotonin reuptake inhibitors (citalopram, escitalopram, paroxetine, sertraline), tricyclic antidepressant (amitriptyline), glutamate antagonist (amantadine), atypical antipsychotics (clozapine, quetiapine), antiseizure medications (carbamazepine, oxcarbazepine, phenytoin, gabapentin, pregabalin, valproate), pure opioid agonists (fentanyl, morphine), bismuth salts, and mood stabilizers (lithium).

The distribution of myoclonus ranges from focal to generalized, even amongst patients using the same drug, which suggests various neuro-anatomical generators. In many cases, DIM subsides with the cessation of the responsible drug, but specialized treatments and therapies are common. Due to the heterogeneous nature of myoclonus secondary to drugs, DIM should always be assessed as a differential diagnosis of myoclonus. The dose-dependent progression of myoclonus from multifocal to generalized and the involvement of CNS symptoms like confusion and visual hallucinations are observed with an increase in the dosage of opioids like fentanyl, indicating the need for the modification of the offending drug’s dose. Pretreatment with certain drugs like alfentanil, dexmedetomidine, propofol, butorphanol, dezocine, and midazolam in etomidate-induced myoclonus emerged as a therapeutic option for suppressing myoclonus. Patients with isolated myoclonus who do not develop any serotonin syndrome or encephalopathy are likely to have a better prognosis. Besides cephalosporins and fluoroquinolones, antibiotics such as lipopeptides, glycopeptides, and tetracyclines were rarely associated with DIM. Certain drugs like TCAs and lithium, when used in combination, are more likely to cause DIM compared to their use in isolation.

In conclusion, our overview confirms the level of evidence supporting various drugs causing myoclonus. Our refinement of DIM may offer clinicians a practical approach to discussing therapeutic benefits and the side effects of drugs, including myoclonus, while maintaining patients’ autonomy and well-being.

## Figures and Tables

**Figure 1 medicina-61-00131-f001:**
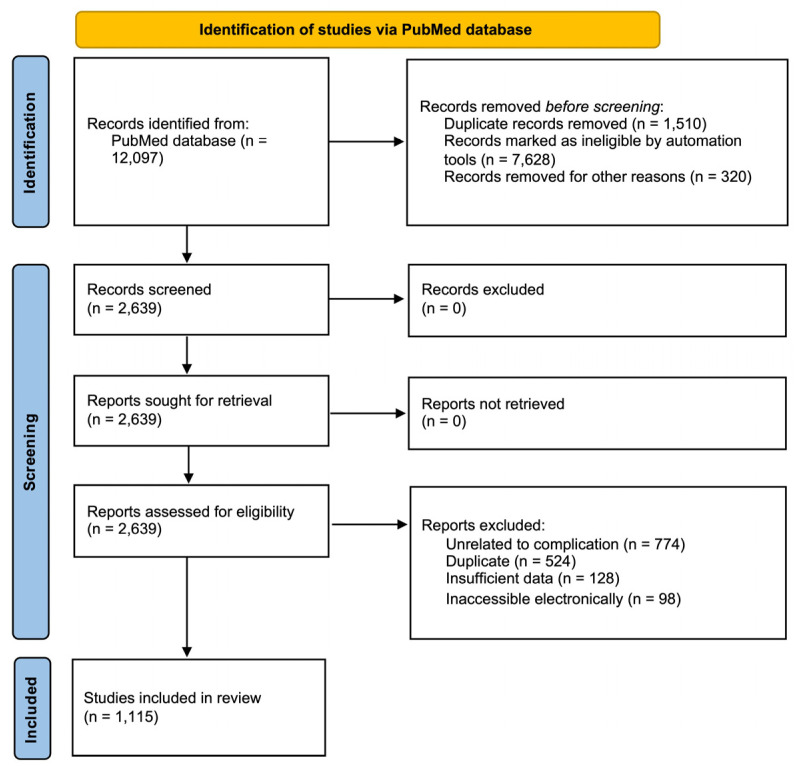
PRISMA flowchart of the screening process.

**Figure 2 medicina-61-00131-f002:**
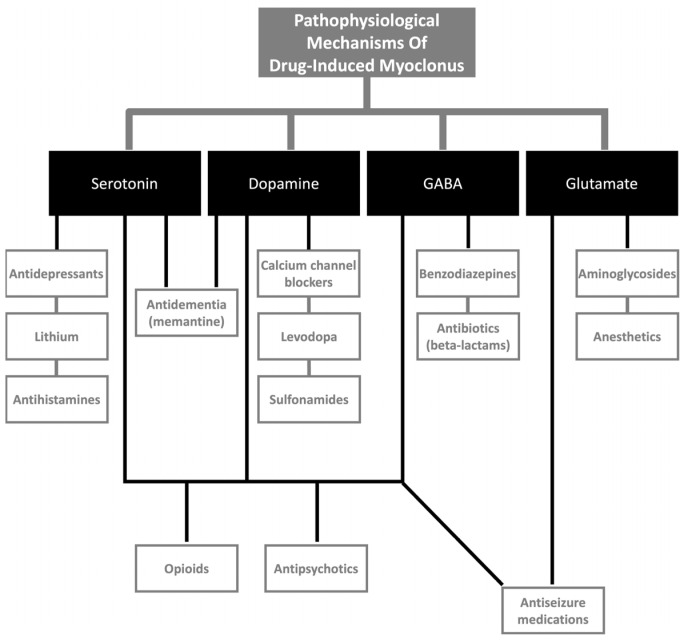
Classification of drug-induced myoclonus based on possible pathophysiological mechanisms.

**Figure 3 medicina-61-00131-f003:**
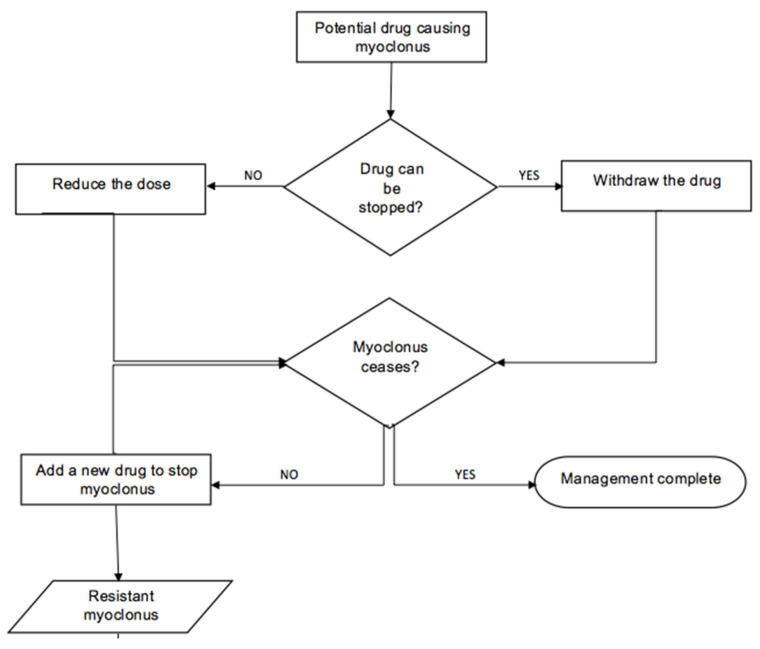
Algorithm of management of drug-induced myoclonus.

**Table 1 medicina-61-00131-t001:** Drug-induced myoclonus and level of evidence.

Class	Drug	Level of Evidence *
Anesthetics	Anesthetic gasses: nitrous oxide	C
	Intravenous anesthetics: phenols (propofol), benzodiazepine (midazolam), opioids (fentanyl), arylcyclohexylamines (ketamine), etomidate	A
	Volatile liquids: enflurane, isoflurane, sevoflurane	B
	Local anesthetics: bupivacaine, dibucaine, lidocaine (lignocaine), prilocaine, tetracaine, levobupivacaine	B
Antibiotics	Penicillins: carbenicillin, penicillin G, oxacillin, amoxicillin-clavulanic acid, nafcillin, piperacillin-tazobactam	B
	Cephalosporins: cefuroxime (2nd generation), cefmetazole (2nd generation), cefotiam (3rd generation), ceftriaxone (3rd generation), ceftazidime (3rd generation), moxalactam (3rd generation), cefepime (4th generation)	A
	Carbapenems: imipenem, meropenem, ertapenem	B
	Fluoroquinolones: ciprofloxacin, moxifloxacin, levofloxacin, gatifloxacin, ofloxacin	A
	Macrolides: erythromycin, azithromycin	C
	Others: aminoglycosides (gentamicin), cotrimoxazole (sulfamethoxazole-trimethoprim), isoniazid, lipopeptides (daptomycin), glycopeptides (vancomycin), tetracyclines (doxycycline), oxazolidinones (linezolid)	C
Antidementia	Cholinesterase inhibitors: donepezil, galantamine, tacrine	C
	Others: memantine	B
Antidepressants	Selective serotonin reuptake inhibitors: citalopram, escitalopram, fluoxetine, fluvoxamine, paroxetine, sertraline	A
	Serotonin-norepinephrine reuptake inhibitors: duloxetine, milnacipran, venlafaxine	B
	Norepinephrine reuptake inhibitors: atomoxetine	C
	Norepinephrine-dopamine reuptake inhibitors: bupropion	B
	Noradrenergic and specific serotonergic antidepressant: mianserin, mirtazapine	C
	Serotonin antagonist and reuptake inhibitors: nefazodone, trazodone	C
	Serotonin modulators and stimulators: vortioxetine	C
	Tricyclic antidepressant: amitriptyline, clomipramine, desipramine, imipramine, nortriptyline	A
	Tetracyclic antidepressant: maprotiline	C
	Monoamine oxidase inhibitors: iproniazid, moclobemide, phenelzine, tranylcypromine	B
	Adjunctive therapy: buspirone	B
Antiemetics	5HT3 antagonists: ondansetron, palonosetron	C
	Dopamine antagonists: domperidone, metoclopramide	B
Antihemorrhagic	Tranexamic acid	C
Antihistamines	H1 antagonist: oxatomide, triprolidine, tripelennamine	C
	H2 antagonist: cimetidine	C
Antineoplastic and immunosuppressive agents	Alkylating agents: busulfan, chlorambucil, cyclophosphamide, ifosfamide	B
	Immunosuppressive agents: cyclosporine, tacrolimus	C
	Monoclonal antibodies: ipilimumab, nivolumab, pembrolizumab	C
	Nitrogen mustards: prednimustine	C
	Nucleoside analogs: 5-fluorouracil, floxuridine, pentostatin (deoxycoformycin)	C
	Topoisomerase inhibitor: irinotecan, pyrazoloacridine	C
Antiparasitic	Antimalarial: mefloquine	C
	Antinematodal: avermectin	C
	Metronidazole	C
Antiparkinsonian	Central anticholinergic: trihexyphenidyl	C
	Catechol-O-methyltransferase inhibitors: entacapone	C
	Dopamine precursor: levodopa	B
	Dopaminergic agonists: bromocriptine, pramipexole	C
	Glutamate antagonist: amantadine	A
	Monoamine oxidase B inhibitors: selegiline	C
Antipsychotics	Typical: haloperidol	B
	Atypical: amisulpride, clozapine, chlorpromazine, olanzapine, perospirone, quetiapine, risperidone, sulpiride, sultopride	A
Antiseizure	Carbamazepine, oxcarbazepine	A
	Clobazam	C
	Lacosamide	C
	Lamotrigine	B
	Phenobarbital	C
	Phenytoin	A
	Gabapentin, pregabalin	A
	Topiramate	B
	Valproate	A
	Vigabatrin	C
Antiviral	Purine analogue: acyclovir, valacyclovir, vidarabine	C
Anxiolytics	Benzodiazepines: lorazepam, midazolam	B
	Others: abecarnil, carisoprodol, tandospirone	C
Cardiovascular	Antianginal: ranolazine	C
	Antiarrhythmics: amiodarone, diltiazem, flecainide, propafenone, verapamil	B
	Angiotensin receptor-neprilysin inhibitor: sacubitril/valsartan	C
	Claudication: buflomedil, naftidrofuryl	C
	Hypertension: amlodipine, carvedilol, enalapril, ketanserin, nifedipine, furosemide	C
	Vasopressors: dobutamine, midodrine	C
Opioids	Pure agonists: dextropropoxyphene, fentanyl, hydrocodone, methadone, morphine, norpethidine, oxycodone, pethidine (meperidine), remifentanil, sufentanil, tramadol	A
	Partial agonists: buprenorphine	C
	Pure antagonists: naloxone, naltrexone	C
Others	Nonsteroidal anti-inflammatory drugs: diclofenac, indomethacin, ketoprofen	C
	Alcohol dependence: acamprosate, disulfiram	C
	Bismuth salts	A
	Cytokine receptor modulators: etanercept, interferon-alpha	C
	Heavy metals: aluminum, lead, magnesium, manganese, nickel	C
	Hormones: corticotropin-releasing hormone, thyrotropin-releasing hormone	C
	Mood stabilizers: lurasidone, lithium	A
	Muscle relaxant: baclofen, gallamine, metaxalone	C
	Organophosphate: aldicarb	C
	Proton pump inhibitors: lansoprazole, omeprazole	C
	Steroids: dexamethasone, prednisolone	C
	Traditional medicine: licorice	C
	Several: atropine, bilimbi fruit, caffeine, bromisoval (bromovalerylurea), bromomethane (methyl bromide), carbon dioxide/monoxide, cobalamin, contrast agent, COVID-19 vaccine, cyclonite, deferoxamine, dextromethorphan, dichloroethane, dieldrin, flumazenil, gasoline, isotretinoin, lindane, metformin, methohexital, mushroom (Sugihiratake), physostigmine, piperazine, pseudoephedrine, Ro5-4864, salbutamol (albuterol), scopolamine, snake bite venom (rattlesnake), sumatriptan, veratramine, zolpidem, zopiclone	C
Recreational drugs	Alcohol, amphetamine, butanone (methyl ethyl ketone), cannabidiol, cathinone, chloralose, cocaine, ecstasy, gamma-butyrolactone, glutethimide, heroin (diamorphine), lysergic acid diethylamide (LSD), methaqualone, methylphenidate, nefopam, strychnine, toluene	C
Animal models	Alphaxalone, bicuculline, catechol, cysteamine, dichloro-diphenyl-trichloroethane (DDT), flurothyl, gadodiamide, gadopentetic acid (gadopentetate dimeglumine), indoleamine, pentylenetetrazole, picrotoxin, pilocarpine, ricinine, tryptophan, urea	D

* Grading level of the evidence: A, more than 20 individuals have been reported to have myoclonus; B, between 5 and 20 individuals have been reported to have myoclonus; C, less than 5 individuals have been reported to have myoclonus; D, only animal studies reporting myoclonus.

**Table 2 medicina-61-00131-t002:** Proposed mechanisms for drug-induced myoclonus.

Medication	Proposed Mechanism	References
Anesthetics	Etomidate-induced myoclonus was correlated with the NMDAR receptor-induced downregulation of potassium-chloride transporter member five protein expression.	Feng et al., (2023) [39]
Antibiotics	Beta-lactam antibiotics selectively antagonize, and quinolones completely inhibit GABA receptors.	Sarva et al., (2012) [81] Post et al., (2004) [73]
	Sulfonamides are associated with abnormalities in dopamine metabolism due to the inhibition of dihydrofolate reductase and increased phenylalanine levels due to the inhibition of phenylalanine metabolism.	Dib et al., (2004) [82] Jundt et al., (2004) [294]
	Aminoglycosides are related to NMDAR receptor activation and excitotoxicity.	Segal et al., (1999) [295]
	Vancomycin likely leads to renal dysfunction, and the cause of myoclonus is related to uremia.	Patel et al., (2018) [84]
Antidementia	Memantine-induced myoclonus might involve altered dopamine, serotonin, and glutamate release levels.	Matsunaga et al., (2001) [296]
Antipsychotics	Serotonergic, dopaminergic, and GABA-ergic mechanisms	Domínguez et al., (2009) [297] Velayudhan et al., (2005) [211] Praharaj et al., (2010). [298]
Benzodiazepine	Likely related to GABAergic transmission. Benzodiazepines were already reported to improve and cause myoclonus.	Valin et al., (1981) [299] Cepeda et al., (1982) [242]
Calcium channel blockers	The mechanism of these effects and the origin of myoclonus are unknown, though Parkinsonism has also been reported with calcium channel blocking agents and attributed to effects on dopamine metabolism.	García-Ruiz et al., (1998) [300]
Levodopa	Serotonergic hypothesis. Anticholinergics, amantadine, and propranolol did not influence the myoclonus. But methysergide improved it.	Klawans et al., (1986) [301]
	Dopaminergic hypothesis. Studies with guinea pigs showed the worsening of myoclonus with pre-treatment with haloperidol and improvement with levodopa or dopamine agonists.	Weiner et al., (1979) [302]
Lithium	It is likely serotonergic activity. Lithium facilitates the presynaptic release of serotonin.	Evidente et al., (1999) [140]
Opioids	The direct neurotoxic effect leads to glutamate receptor hyperexcitability, glycine-mediated disinhibition, antagonism of GABAergic activity, and serotonergic and dopaminergic pathways.	Han et al., (2002) [260]
Serotonergic drugs	Myoclonus worsened with 5-HTP and valproate and improved with methysergide. Also, serotonin reuptake inhibitors can cause isolated myoclonus or myoclonus as a part of serotonin syndrome.	Giménez-Roldán et al., (1988) [303] Feighner et al., (1990) [304] Sternbach et al., (1991) [305]
	The interaction between serotonin 5-HT1A and 5-HT2 receptors seems necessary to induce myoclonus since selective agonists for 5-HT1A and 5-HT2 receptors do not induce myoclonus when given individually.	Eison et al., (1993) [306] Pappert et al., (1998). [307]
Tricyclic antidepressants	Serotonergic hypothesis. A combination of 5-HTP and imipramine showed myoclonus, which did not improve with the antagonism of norepinephrine, dopamine, and acetylcholine receptors.	Klawans et al., (1986) [301]

Abbreviations: GABA, gamma-aminobutyric acid; NMDAR, N-methyl-D-aspartate receptor; 5-HTP, 5-hydroxytryptophan.

**Table 3 medicina-61-00131-t003:** Proposed classification for drug-induced myoclonus.

Type	Main Mechanism	Definition	Examples	Reference
Type 1	Serotonin syndrome	Fulfillment of Sternbach’s or Hunter’s criteria	Moclobemide and pethidine	Gillman et al., (1995) [133]
			Trazodone and buspirone	Goldberg et al., (1992) [238]
			Tramadol and iproniazid	Larquier et al., (1999) [132]
Type 2	(2A) Hepatic encephalopathy	Concurrent hepatic encephalopathy and myoclonus, myoclonus likely unrelated to indirect liver injury	Valproate	Rissardo et al., (2021) [231]
			Phenytoin	Rissardo et al., (2022) [226]
			Carbamazepine	Risssardo et al., (2020) [218]
	(2B) Non-hepatic encephalopathy	Non-hepatic encephalopathy and myoclonus, also known as Creutzfeldt–Jakob-like syndrome.	Bismuth	Borbinha et al., (2019) [286]
			Lithium	Rissardo et al., (2022) [289]
			Amitriptyline	Rissardo et al., (2020) [111]
Type 3	Unknown	All the other patients do not have type 1 or type 2 features. Interestingly, this group of individuals likely involves poorly understood mechanisms.	Etomidate	Doenicke et al., (1999) [38]
			Benzodiazepines	Magny et al., (1994) [23]
			Amantadine	Rissardo et al., (2023) [173]

## Data Availability

No new data were created or analyzed in this study.

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
