# Peer review of "Drug-Induced Myoclonus: A Systematic Review"

_medicina, 2025, doi:10.3390/medicina61010131_

Round 1
Reviewer 1 Report
Comments and Suggestions for Authors
The submitted review entitled: Drug-Induced Myoclonus: A Systematic Review, presents an extensive study that confirms the level of evidence supporting various drugs causing myoclonus (DIM). The following minor issues should be addressed before further steps:
Abstract
-Line 13: change was to is
-Line 14: change con-ditions to conditions
-Line 16: The following is not clear ( Two reviewers identified and as)?
-Line 31: explain whether the following statement is derived from this study; or this study confirms it ( DIM can be categorized…)
Results:
-Line 122: including the term (animal models of myoclonus) is a clever step leading to more comprehensive picture of the DIM.
-Table 2, 3 and 5: it is suggested to put drug class/category on a new left column to avoid confusion (as in Table 4, which looks better).
-Figure 2: complete the first box to be: Pathophysiological mechanisms of DIM.
Author Response
Reviewer 1
The submitted review entitled: Drug-Induced Myoclonus: A Systematic Review, presents an extensive study that confirms the level of evidence supporting various drugs causing myoclonus (DIM). The following minor issues should be addressed before further steps:
Abstract
-Line 13: change was to is
Authors: addressed.
-Line 14: change con-ditions to conditions
Authors: addressed.
-Line 16: The following is not clear ( Two reviewers identified and as)?
Authors: addressed.
-Line 31: explain whether the following statement is derived from this study; or this study confirms it ( DIM can be categorized…)
Authors: addressed.
Results:
-Line 122: including the term (animal models of myoclonus) is a clever step leading to more comprehensive picture of the DIM.
Authors: We appreciate the reviewer comment.
-Table 2, 3 and 5: it is suggested to put drug class/category on a new left column to avoid confusion (as in Table 4, which looks better).
Authors: addressed.
-Figure 2: complete the first box to be: Pathophysiological mechanisms of DIM.
Authors: addressed.
We appreciate the reviewer comments and we believe that this comments significantly improved the quality of the manuscript regarding understanding of the findings.
Reviewer 2 Report
Comments and Suggestions for Authors
Drug-induced myoclonus is a very serious complication, and therefore the topic of the manuscript is undoubtedly relevant. However, it makes sense for the authors to describe in more detail the various mechanisms of myoclonus in the Introduction section. Due to the fact that the manuscript is large, including due to the large number of tables, it is recommended to present Summary Table No. 1 as a figure or diagram, since the data in it are duplicated with the tables given later in the manuscript. Also, other tables that have only 2 columns, the first of which indicates the name of the drug, and the second - the year and the authors' names, should also be presented as a text paragraph. At the same time, tables that present data on the mechanism of myoclonus are undoubtedly valuable. In the Conclusion section, the authors should describe and analyze the results in a more structured way with possible recommendations for practitioners on the use of drugs that cause myoclonus.
Author Response
Reviewer 2
Drug-induced myoclonus is a very serious complication, and therefore the topic of the manuscript is undoubtedly relevant. However, it makes sense for the authors to describe in more detail the various mechanisms of myoclonus in the Introduction section. Due to the fact that the manuscript is large, including due to the large number of tables, it is recommended to present Summary Table No. 1 as a figure or diagram, since the data in it are duplicated with the tables given later in the manuscript. Also, other tables that have only 2 columns, the first of which indicates the name of the drug, and the second - the year and the authors' names, should also be presented as a text paragraph. At the same time, tables that present data on the mechanism of myoclonus are undoubtedly valuable. In the Conclusion section, the authors should describe and analyze the results in a more structured way with possible recommendations for practitioners on the use of drugs that cause myoclonus.
Authors:
Dear Reviewer, we appreciate your comments regarding out manuscript.
We would like to maintain the current introduction type. The idea with the introduction is to provide a brief overview of myoclonus definition, pathophysiology, and causes, specifying drug-induced myoclonus to those authors that are not familiar with this term. The idea of placing the pathophysiology it would be great, but this is long term discussion and it takes at least some pages to explain the recent discoveries regarding the neurophysiology of myoclonus.
Regarding table being converted to figures. We already though in transforming the table as figure. However, we will losing a significant point of this manuscript. The current manuscript is supposed to be like a pilar or more like a searching database for drug-induced myoclonus. For example, if a person searches in Google, Google Scholar, or PubMed, we want for out manuscript to be the first to appear, but for this to happen need to be tables, if we transform to figures we will lose this.
Regarding the tables, the other reviewer liked and give a great idea of transforming in three column tables. The reason for reference being write at the right-side is the fact to be like a proof that this was already reported and the reader can go beyond and search for that author report or article regarding the myoclonus case. Therefore, we would like to maintain these tables as they are at the present moment.
Regarding the conclusion, at the present moment the conclusion provides a summary of the results found like level A evidence, followed by the myoclonus phenomenology, and some specific findings that we found, and we lastly described the classification that we are proposing. The idea of providing a more specific management is great, but the main author is already studying drug-induced movement disorder for the last five years. There are some key factors regarding management that should be pointed out. A common sense is that a drug causing myoclonus may differ from other drug regarding mechanism, so not every drug-induced myoclonus has the same management. Therefore, myoclonus is a drug-induced movement disorder easy to understand because we can evaluate frequency and amplitude of movement based on the management. After all of this, specific myoclonus secondary to medications should have specific guidelines, or we should only discontinue the medication. This is beyond the scope of the current review, and it is something that will be likely changing in the future about the definition of movement disorders and their management; especially regarding drug-induced movement disorders, which we still have some specialists believing that the cases in the literature are psychogenic disorders.